# The effect of metacognitive training on confidence and strategic reminder setting

Nicole C. Engeler [1,2]*, Sam J. Gilbert[1]

**1** UCL Institute of Cognitive Neuroscience, London, United Kingdom, **2** UCL Department of Experimental Psychology, London, United Kingdom

* nicole.engeler.19@ucl.ac.uk

**Data Availability Statement:** All data files and analysis codes will be held in a public repository, available at https://osf.io/ebp4z/.

**Funding:** Funding for this study was received by SJG from the Economic and Social Research

## Abstract

Individuals often choose between remembering information using their own memory ability versus using external resources to reduce cognitive demand (i.e. 'cognitive offloading'). For example, to remember a future appointment an individual could choose to set a smartphone reminder or depend on their unaided memory ability. Previous studies investigating strategic reminder setting found that participants set more reminders than would be optimal, and this bias towards reminder-setting was predicted by metacognitive underconfidence in unaided memory ability. Due to the link between underconfidence in memory ability and excessive reminder setting, the aim of the current study was to investigate whether metacognitive training is an effective intervention to a) improve metacognitive judgment accuracy, and b) reduce bias in strategic offloading behaviour. Participants either received metacognitive training which involved making performance predictions and receiving feedback on judgment accuracy, or were part of a control group. As predicted, metacognitive training increased judgment accuracy: participants in the control group were significantly underconfident in their memory ability, whereas the experimental group showed no significant metacognitive bias. However, contrary to predictions, both experimental and control groups were significantly biased toward reminder-setting, and did not differ significantly. Therefore, reducing metacognitive bias was not sufficient to eliminate the bias towards reminders. We suggest that the reminder bias likely results in part from erroneous metacognitive evaluations, but that other factors such as a preference to avoid cognitive effort may also be relevant. Finding interventions to mitigate these factors could improve the adaptive use of external resources.

## Introduction

In our daily lives we must frequently remember to execute intentions such as buying ingredients for a meal or attending future appointments. However, our unaided memory abilities are limited [1]. Hence, we frequently choose to enhance our memory for delayed intentions with external tools, for instance by taking notes or by setting reminders in our smartphones [2, 3]. Reducing the cognitive demands of a task in this way has been termed cognitive offloading, and creating external triggers in order to remember delayed intentions is known as intention

Council, ES/N018621/1, https://esrc.ukri.org.
Funders played no role in the study design, data
collection and analysis, decision to publish, or
preparation of the manuscript.

**Competing interests:** The authors have declared
that no competing interests exist.

offloading [4]. With the development of technologies such as smartphones and other smart devices, the use of technology to lessen an individual's memory load has become increasingly ingrained in the completion of everyday tasks [5–7]. However, using reminders involves both costs (e.g. the time and effort creating them) and benefits (e.g. increased likelihood of remembering), and individuals often need to evaluate whether it would be more beneficial for them to create a reminder or not. Previous studies have suggested that an accurate estimation of our unaided memory abilities may be necessary to make optimal choices in reminder-setting [8–12]. Therefore, this study examines an intervention designed to a) improve the accuracy of participants' self-judgments, and b) reduce bias in offloading decisions.

To study cognitive offloading in memory, Gilbert [9] developed an intention offloading task in which participants dragged numbered circles to the bottom of a box in sequential order. During this ongoing task, participants also completed delayed intentions in which they were required to drag target circles to alternative locations (left, right or top) of the box. Dragging a sequence of numbered circles out of the box completed a 'trial'. As an alternative strategy to relying on their own memory and mentally rehearsing delayed intentions, participants could 'offload' the need to remember by dragging target circles to the alternative locations at the beginning of each trial, before they were reached in the sequence. The target circles could then act as reminders, akin to someone leaving an object by the front door so that they remember to take it when leaving the house the next day. This study found that setting external reminders (i.e. offloading) improved performance. Additionally, participants set these reminders adaptively, based on the internal cognitive demands of the task (i.e. the number of items to remember and the presence of distractions). These findings suggest that individuals decide whether to set reminders based on a metacognitive evaluation of the difficulty of the task. In line with this, findings from multiple studies point toward metacognitive judgments as a key factor in decisions about offloading. Using the same paradigm as detailed above, Gilbert [10] showed that choosing to set external reminders was predicted by participants' metacognitive confidence, as people with lower confidence in their memory ability used more reminders, even when that confidence was unrelated to objective performance. The relationship between confidence and intention offloading was replicated by Boldt and Gilbert [8], both when reminder-setting was instructed and when it was spontaneously generated. Similarly, in another study involving the recall of word pairs, lower confidence in memory ability was linked to more frequent requests for hints, even when performance was controlled for [12]. In line with this, a survey study showed a negative correlation between self-reported internal memory ability and the use of memory offloading [13]. These results suggest that decisions of whether to set reminders are influenced by potentially erroneous metacognitive evaluations of internal memory abilities.

To investigate whether participants weigh costs and benefits of reminder-setting optimally or whether they show systematic bias, Gilbert et al. [11] adapted the paradigm by Gilbert [9]. Participants could either earn a maximum reward (10 points) for correctly remembered target circles when using their own memory, or earn a lesser reward (between 1–9 points) when setting reminders to increase the number of circles remembered. All experiments found a bias toward the use of reminders, predicted by participants' inaccurate metacognitive underconfidence in their own internal memory abilities. Furthermore, metacognitive interventions have been shown to influence the reminder bias. In an experiment where one group of participants received metacognitive advice about whether they would be likely to score more points using their own memory or reminders, the reminder bias was eliminated [11, Experiment 2]. Another experiment [11, Experiment 3] manipulated feedback valence (positive or negative) and the difficulty of practice trials (easy or hard). This produced one group (difficult practice, negative feedback) which was significantly underconfident, and another group (easy practice,

positive feedback) that was significantly overconfident. Both groups used reminders significantly more often than would have been optimal. Seeing as a bias towards reminders can be observed both in the context of under- and over-confidence, it seems likely that metacognitive bias can partially, but not fully, explain the reminder bias.

Metacognitive bias does not occur in the same way for all tasks, populations and individuals: some experiments find that people are overconfident in their memory ability, but other experiments find that people are underconfident [10, 14–18]. Therefore, we aim to create a metacognitive intervention which can potentially remedy biases in either direction. Asking participants to make predictions about their own performance, and providing feedback on the accuracy of those predictions, may be a suitable method of "training" metacognitive accuracy. Indeed, multiple studies examining the role of feedback on participants' judgments have found a reduction in metacognitive bias and improvements in judgment accuracy [19–21].

Hence, the primary aim of this study is to investigate whether metacognitive training, i.e. providing participants with feedback on their metacognitive judgments, is an effective intervention to a) improve participants' metacognitive judgment accuracy, and b) reduce bias in offloading behaviour. We predict that participants in the experimental condition will make more accurate judgments than participants in a control group. Moreover, we predict that an improvement in metacognitive accuracy may result in more optimal strategy choices, as measured by the reminder bias. As individuals rely on offloading to organise their behaviour, but do not always offload optimally, finding interventions to influence individuals' offloading strategies could improve behavioural organisation in everyday life.

## Methods

To view a demonstration of the experimental task, please visit [http://samgilbert.net/demos/NE1/start.html]. This demonstration version omits the information and consent pages, informs the visitor at the beginning whether they have been randomised to the feedback or no-feedback control condition, and does not record any participant data. Other than this, the demonstration version of the task is identical to the one undertaken by the actual participants in this study.

### Design

The present study adapted the paradigm developed by Gilbert et al. [11] to investigate whether metacognitive training has an impact on a) metacognitive judgment accuracy, and b) strategic reminder setting. During 'metacognitive training' participants were asked to make pre-trial predictions about their performance, and then received feedback on their judgment accuracy post-trial. Using a between-subjects design, participants were randomly allocated to either an experimental condition with metacognitive feedback training or a control group without. The experimental manipulation occurred during 'forced trials' only (described below), with all else being identical between groups. Before commencing data collection, all hypotheses, experimental procedures, and analysis plans were pre-registered [https://osf.io/ebp4z/].

### Optimal reminders task

See Fig 1 for a schematic illustration of the task. Participants were presented with six yellow circles randomly positioned within a box, on their device screen. Each circle contained a number which participants were asked to drag sequentially (1, 2, 3 etc.) to the bottom of the box. When a circle was dragged out of the box, a new circle would appear in its vacated location, continuing the next number in sequence (i.e. if 1–6 were on screen, 7 would appear in the location of 1 after it got dragged out). Each trial contained 25 circles presented in sequence. Sometimes, new circles first appeared in another colour (blue, orange or pink) rather than yellow,

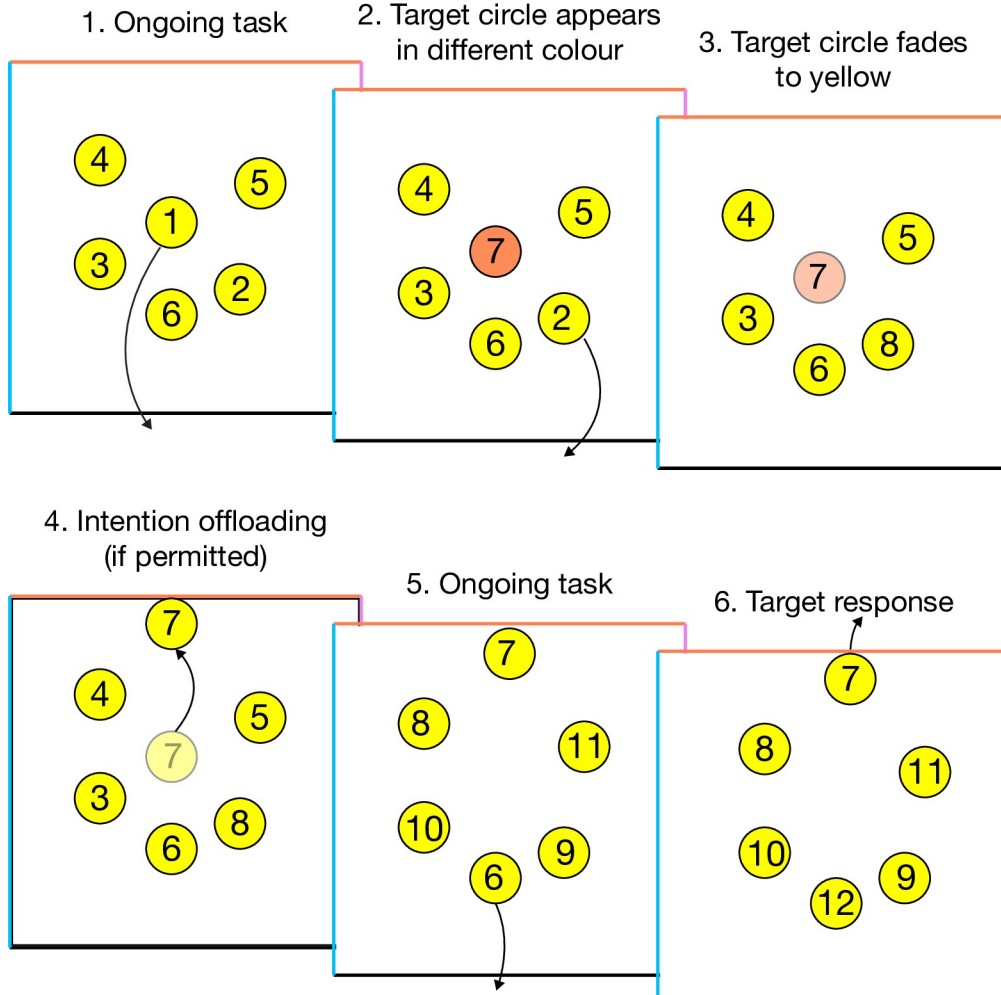

**A. Sequence of events within a trial**

1. Ongoing task
2. Target circle appears in different colour
3. Target circle fades to yellow
4. Intention offloading (if permitted)
5. Ongoing task
6. Target response

**B. Example stimulus display prior to a choice trial**

You have scored a total of 100 points so far.

This time you have a choice.

Please touch the option that you prefer:

Special circles worth **6** points

Reminders allowed

Special circles worth **10** points

Reminders **not** allowed

**Fig 1. Schematic illustration of the optimal reminders task.**

but after 2 seconds those circles faded to yellow as well. These were target circles which corresponded with the alternative sides of the box (left, top and bottom). Hence, a circle initially appearing in an alternative colour meant that the circle should eventually be dragged to its corresponding side of the box once reached in the sequence. For instance, if after dragging number 1 to the bottom, 7 initially appeared in blue, the participant had to drag 2–6 to the bottom before dragging 7 to the left (blue) side. When a target circle was dragged to the correct side of the box it turned green before disappearing. Circles dragged wrongly to an alternative side of the box turned red before disappearing.

To remember to drag initially nonyellow circles to alternative locations when reached in the sequence, participants had to form a delayed intention. There were two strategies for remembering these intentions: participants could either rely on their internal (unaided) memory or create an external reminder. To create external reminders, target circles had to be dragged near the instructed alternative location as soon as the circle appeared on screen (because circles faded back to yellow quickly). Once the target circle was reached in a sequence, its location would remind participants of their intention. There were 10 target circles per trial, which always appeared between positions 7 and 25, distributed as evenly as possible. Seeing as participants had to remember multiple intentions at once, it was unlikely that they would remember all of them if they relied on their internal memory. However, the task was easier when external reminders were used.

In the optimal reminders paradigm participants completed trials in which they were forced to use their internal (unaided) memory, as well as trials in which they had to set external reminders. The paradigm also contained choice trials in which participants had to decide between receiving the maximum reward for each remembered target circle when using their own memory, or receiving a smaller reward when using reminders. Correct target circles were always worth 10 points when the internal strategy was chosen, and varied between 1–9 points for the external strategy. Assigned point values remained for an entire trial. During the nine choice trials, the values for reminders (1–9) were presented in a randomised order. Participants were instructed to choose the strategy with which they believed they could score the most points. To do so, participants had to consider both the amount of points they would receive per correct circle, and the number of circles they thought they were likely to remember correctly with each strategy. Additionally, including forced internal and external trials allowed us to determine each participant's optimal indifference point, i.e. the point at which they should be indifferent between the two strategies, based on their accuracy during the forced trials. This indifference point could then be compared to an individual's actual indifference point as seen from the decisions made during choice trials. The difference between a participants' optimal and actual indifference point is the reminder bias (see data analysis section).

## Participants

As specified in our preregistration, we aimed for a final sample size of 116. Our power calculation was based on an experiment which, like the present study, tried to influence metacognitive judgments and strategy choices using Gilbert et al.'s [11] paradigm. In their experiment a group of participants received metacognitive advice about which strategy to use before choosing a strategy option [11, Experiment 2]. The reminder bias was eliminated in this group, and it was significantly reduced in comparison with a control group who did not receive advice (Cohen's $d$ = .55). Assuming that the influence of metacognitive feedback may be comparable to the influence of metacognitive advice, and based on a desired power of 90%, this yielded a required sample of 116 (58 participants in each group) to conduct the between-subject comparisons (G*Power 3.1). A total of 133 participants were tested to reach the planned sample

size of 116, after applying the preregistered exclusion criteria. These criteria were designed to ensure that included participants engaged with the task as intended. Participants were excluded for a) having higher accuracy for forced internal (own memory) than forced external (with reminders) trials (n = 4), b) lower than 70% accuracy during external trials (n = 6), c) a negative correlation between target value and likelihood of choosing to set reminders, which suggests random or counter-rational strategy choices (n = 4) and d) a metacognitive bias score more than 2.5 standard deviations from the group mean (n = 3). The final sample had a mean self-reported age of 37 (*SD* = 11.13, range: 20–71), with 64 females, 41 males and one other. All participants provided informed consent before participating and the research was approved by the UCL Research Ethics Committee.

## Procedure

Participants were recruited from the Amazon Mechanical Turk website and completed the experiment on their computer, accessing it via a provided weblink. Participation was restricted to individuals with a minimum of 90% Mechanical Turk approval rate, and to those reporting a location in the U.S. in order to reduce heterogeneity and to remain consistent with previous studies. The median duration to complete the experiment was 31 minutes. Participants were paid $7.50 for taking part.

See Fig 2 for a visualization of the task procedure. Participants first performed a practice session which required them to respond accurately to a target circle in order to proceed. This ensured that they understood the task instructions properly. Following this, participants in both groups completed 5 forced internal and 4 forced external trials in alternating order, beginning and ending with an internal trial. This served two purposes. First, accuracies in the two conditions could be used to calculate the optimal indifference point, i.e. the number of points offered for each target when using reminders at which an unbiased individual would be indifferent between the two strategies (see data analysis section). Second, this provided the opportunity for the metacognitive training group to give performance predictions and to receive feedback on those predictions. The reason for including more internal than external trials was so that the number of trials in this phase was matched to the subsequent choice phase. Furthermore, it was of more interest to train participants' metacognitive accuracy during internal trials than external trials, as the reminder bias has been previously linked to erroneous underconfidence in unaided internal memory ability [11]. Prior to each trial, participants in both groups were informed of the number of points they had scored so far and were told which strategy they had to use in the upcoming trial. Participants in the experimental feedback group were additionally instructed to provide performance predictions before they began each trial. For this, participants had to use a moveable slider on their screen to indicate what percentage of target circles (0% - 100%) they thought they would be able to correctly drag to the instructed side of the square during the next trial. After each trial, participants in the experimental group received feedback about their judgment accuracy: they were reminded of

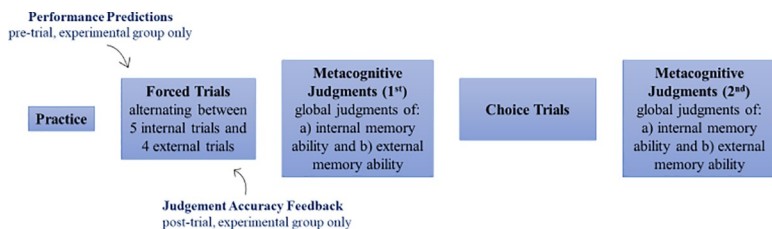

**Fig 2. Summary of the task procedure.**

their predicted accuracy, informed of their actual accuracy, and told whether they had under-estimated, overestimated or accurately estimated their memory ability. Participants in the control group did not make performance predictions or receive feedback. After completing the forced internal and forced external trials, participants in both groups were asked to make metacognitive evaluations of their accuracy in the internal and external conditions: *"Please use the scale below to indicate what percentage of the special circles you will correctly drag to the instructed side of the square, on average. 100% would mean that you always get every single one correct. 0% would mean that you can never get any of them correct"*. For both global judgments, participants used a moveable slider to select any value between 0–100%. This allowed us to investigate whether metacognitive training improved the subsequent accuracy of metacognitive predictions. Next, participants received instructions about choice trials and subsequently completed these. Before each trial participants were informed of the total number of points they had scored so far. Finally, participants in both groups were again asked to judge their internal memory ability, as well as their ability whilst using reminders, on a 0–100% scale, with the following wording: *"You have now finished doing the task. But we would like you to make some more predictions. Suppose you had to do the task again using your **own** memory.* [or "using **reminders**", depending on which judgement]. *What percentage of the special circles do you think you would be able to correctly drag to the instructed side of the square, on average? 100% would mean that you always get every single one correct. 0% would mean that you could never get any of them correct. Please remember that you should just answer about your ability to do the task with your **own memory*** [or: "with **reminders**", depending on which condition]*"*. This allowed us to examine whether any effect of metacognitive training on metacognitive evaluations was maintained at the end.

## Data analysis

Data were analysed using R version 4.0.0 and RStudio Version 1.3.959. T-tests did not assume equal variances and degrees of freedom were adjusted accordingly. Data and code to reproduce the analyses below can be downloaded from [https://osf.io/ebp4z/]. All statistical analysis was conducted as outlined in our preregistered plan. Measures and calculations are based on those by Gilbert et al. [11] and are described below:

1. Optimal Indifference Point (OIP): This is the reminder value (1–9) at which an unbiased individual should be indifferent between the internal and the external strategy option, based on their mean target accuracy (i.e. the mean number of correct target circles) on forced internal trials ($ACC_{FI}$), and the mean target accuracy on forced external trials ($ACC_{FE}$). The optimal indifference point is calculated as: $OIP = (10 \times ACC_{FI}) / ACC_{FE.}$

2. Actual Indifference Point (AIP): This is the point at which participants are actually indifferent to the two strategy choices and are equally as likely to choose either option. This was calculated by fitting a sigmoid curve to the strategy choices across the 9 reminder target values (1–9), using the R package 'quickpsy' bounded to the range 1–9.

3. Reminder Bias: The reminder bias is the difference between the optimal and the actual indifference point (OIP–AIP). If participants were unbiased between the two strategy options, the actual and optimal indifference points would match. A positive value indicates that a participant is biased toward using more reminders than is optimal.

4. Metacognitive Judgments: Participants gave four global metacognitive judgments: two internal and two external judgment responses, given before the experimental choice trials began and after all trials were completed.

5. Metacognitive Bias: The metacognitive bias score indicates the difference between participants metacognitive judgements and their objective accuracy levels (i.e. the mean accuracy for

the forced internal and forced external trials). A positive number indicates overconfidence, and a negative value indicates underconfidence.

## Results

Accuracy in the forced internal trials (feedback group: $M$ = 56.17%, $SD$ = 16.76; control group: $M$ = 50.24%, $SD$ = 14.11) was lower than accuracy in the forced external trials (feedback group: $M$ = 95.91%, $SD$ = 5.61; control group: $M$ = 96.29%, $SD$ = 4.92).

In order to assess the accuracy of participants' metacognitive judgments and investigate whether metacognitive training improves judgment accuracy, two metacognitive bias scores (the internal bias and the external bias) were initially calculated by averaging the first and second global judgments. This is in order to avoid type-1 error (see below for additional analyses investigating whether there was an effect of timepoint). One sample t-tests revealed that participants in the control group were significantly underconfident in both their internal ($t$(57) = -6.83, $p$ < .0001, $d$ = -.90) and external memory abilities ($t$(57) = -5.16, $p$ < .0001, $d$ = -.68), whereas participants in the feedback group did not display any significant metacognitive bias (internal bias: $t$(57) = -.34, $p$ = .74, $d$ = -.045; external bias: $t$(57) = -1.91, $p$ = .061, $d$ = -0.25). See Fig 3 for a visualisation of these results. Direct comparisons showed that metacognitive bias was significantly different between the two groups (internal bias: $t$(111.55) = 4.2, $p$ < .0001, $d$ = .78; external bias: $t$(103.38) = 3.08, $p$ = .0013, $d$ = .57); note that these $p$ values are for a one-tailed independent samples t test, in accordance with our preregistered plan. Group differences in judgments of both internal and external memory ability were also observed when raw metacognitive judgments rather than metacognitive bias scores were investigated (internal judgment: $t$(113.58) = 4.63, $p$ < .0001, $d$ = .86; external judgment: $t$(100.08) = 2.49, $p$ = .0072, $d$ = .46); note that these $p$ values are for one-tailed tests, in accordance with our preregistered plan. Furthermore, a mixed ANOVA was conducted to evaluate whether metacognitive bias scores differed according to timepoint (first judgment, second judgment) or condition (internal, external), and whether these effects were modulated by group. The test produced a main effect of group on metacognitive bias scores, F(1,114) = 23.74, $p$ < .0001, $\eta_p^2$ = .17. There was also a main effect of condition (F(1, 114) = 5.43, $p$ = .022, $\eta_p^2$ = .045), as metacognitive underconfidence was more pronounced in the internal ($M$ = -7.74, $SD$ = 18.91) than the external condition ($M$ = -3.75, $SD$ = 7.89), but there was no significant main effect of timepoint

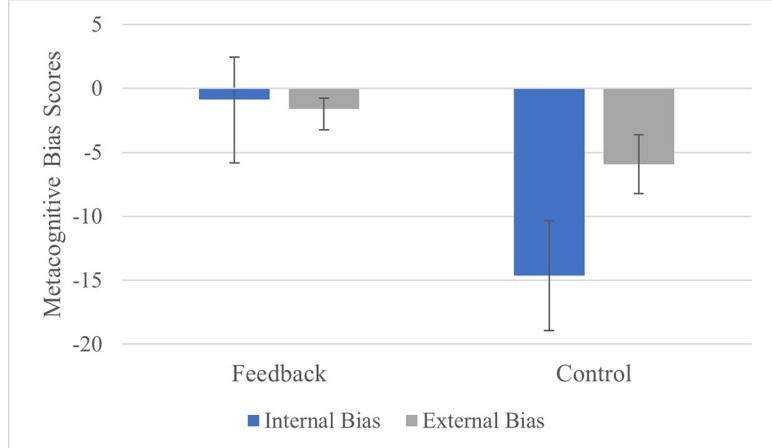

**Fig 3. Internal and external metacognitive bias scores for the feedback and the control group.** Error bars represent 95% confidence intervals.

(F(1,114) = .038, $p$ = .85, $\eta_p^2$ = .00034). These main effects were qualified by a significant group x condition interaction (F(1,114) = 7.61, $p$ = .007, $\eta_p^2$ = .063) with a larger effect of metacognitive feedback on internal bias (feedback: $M$ = -0.84, $SD$ = 18.94; control: $M$ = -14.63, $SD$ = 16.31) than external bias (feedback: $M$ = -1.58, $SD$ = 6.28; control: $M$ = -5.93, $SD$ = 8.75). There was also a group x time interaction (F(1, 114) = 9.52, $p$ = .003, $\eta_p^2$ = .077), reflecting a negative shift in bias from time 1 ($M$ = 0.35, $SD$ = 13.8) to time 2 ($M$ = -2.77, $SD$ = 10.2) in the feedback group, but a positive shift from time 1 ($M$ = -12.05, $SD$ = 11.64) to time 2 ($M$ = -8.51, $SD$ = 10.32) in the control group. The three-way interaction was also significant (F(1,114) = 4.69, $p$ = .032, $\eta_p^2$ = .04); see Table 1 for means and standard deviations. One-tailed independent samples t-tests showed that group differences in metacognitive bias scores were found regardless of timepoint or condition (internal condition, first judgement: $p$ < .0001, d = .84; internal condition, second judgement: $p$ = .0075, d = .46; external condition, first judgement: $p$ = .0014, d = .57; external condition, second judgement: $p$ = .014, d = .41).

Turning to the question of whether participants exhibit a reminder bias and whether this bias is influenced by metacognitive training, one-tailed one sample t-tests revealed that participants in both groups were significantly biased towards setting more external reminders than was optimal (feedback group: $t(57)$ = 4.11, $p$ < .0001, d = 0.54; control group: $t(57)$ = 3.43, $p$ = .0006, d = 0.45). See Fig 4 for a visualisation of these results. Contrary to our prediction, the reminder bias was numerically larger for the feedback group ($M$ = 1.64, $SD$ = 3.04) than the control group ($M$ = 1.16, $SD$ = 2.57). Seeing as our pre-registered plan was to investigate any difference in the opposite direction with a one-tailed test, it is not appropriate to conduct any further statistical analysis of the observed effect.

To further investigate whether the reminder bias is related to metacognitive bias, we calculated the Pearson correlation between the reminder bias and a) internal metacognitive bias, and b) external metacognitive bias, for each group separately. For participants in the feedback group there was no significant correlation between metacognitive biases and reminder bias scores (internal bias: $r(56)$ = -0.004, $p$ = .98; external bias: $r(56)$ = .15, $p$ = .26). Although there was no significant correlation between external bias and reminder bias ($r(56)$ = -.095, $p$ = .48) in the control group either, a significant correlation between participants' internal metacognitive bias and their reminder bias was found ($r(56)$ = -.31, $p$ = .018): the more underconfidence was displayed, the higher the bias for reminders, as has been previously found [11]. This suggests that the relationship between internal metacognitive bias and reminder bias was specific to the no-feedback group. However, we note that a direct comparison between the correlation coefficients in the two groups (based on Fisher's r-t-z transformation) did not yield a significant effect (z = 1.17, p = .24). Therefore, no strong conclusions about specificity may be drawn. See Fig 5 for scatterplots depicting the relationship between internal metacognitive bias and reminder bias scores in the feedback group and the control group, respectively.

**Table 1. Means and standard deviations for the three-way interaction of group x condition x time.**

|  | Internal Bias | | External Bias | |
|  | Time 1 | Time 2 | Time 1 | Time 2 |
|---|---|---|---|---|
| Feedback |  |  |  |  |
| Mean | 3.29 | -4.98 | -2.60 | -0.56 |
| SD | 26.47 | 17.30 | 5.91 | 8.31 |
| Control |  |  |  |  |
| Mean | -16.10 | -13.20 | -8.00 | -3.86 |
| SD | 19.11 | 18.30 | 12.00 | 7.76 |

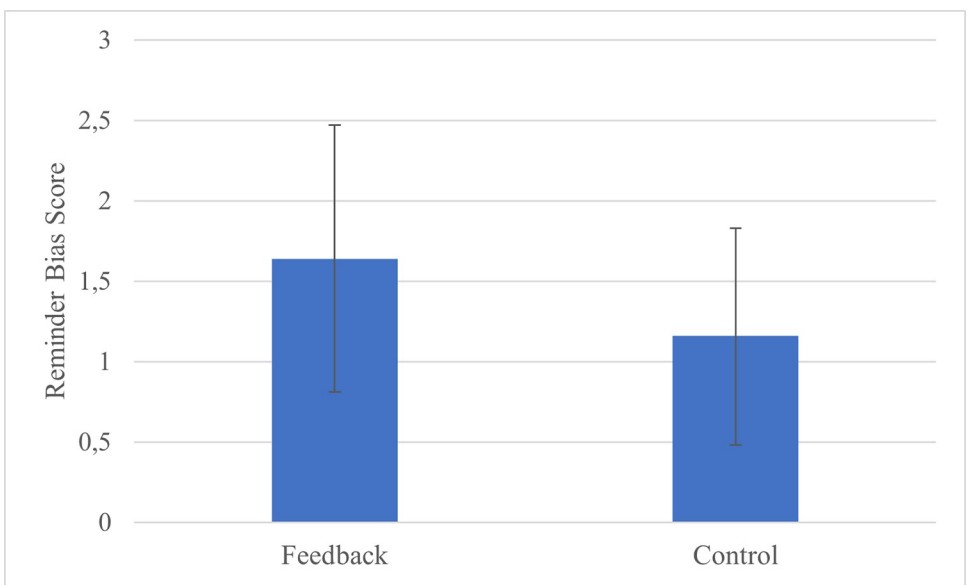

**Fig 4. Reminder bias scores for the feedback and the control group.** Error bars represent 95% confidence intervals.

## Discussion

Enhancing our internal memory ability by using cognitive tools involves both costs and benefits. Previous studies [8–12] have suggested that we do not accurately assess our memory ability and thus do not make the most optimal decisions about whether or not to use external reminders. Using the optimal reminders paradigm, we investigated whether metacognitive training, i.e. providing participants with feedback on their metacognitive judgments, is an effective intervention to a) improve metacognitive judgment accuracy, and b) whether this leads to more optimal offloading behaviour, as measured by the reminder bias.

In line with previous studies [11, 12, 22], using reminders improved task performance, demonstrating the benefit of using external tools in aiding our memory. Further, participants in the control group were significantly underconfident in both their internal memory ability as

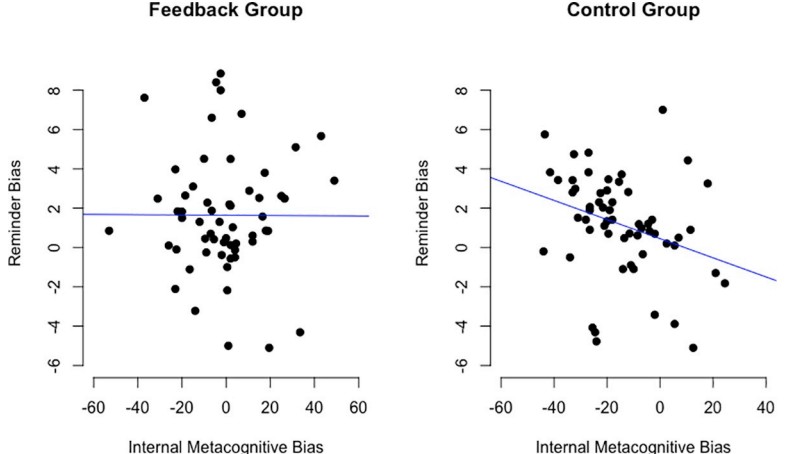

**Fig 5. Scatterplot depicting the correlation between internal metacognitive bias and reminder bias scores in the feedback and control groups, with a line of best fit.**

well as their memory ability when setting reminders. However, they underestimated their internal memory ability to a larger degree than their memory ability with reminders. In comparison, participants who received metacognitive feedback displayed neither internal nor external metacognitive bias, as they made more accurate judgments about their memory performance for both memory strategies. Group differences were significant for both internal and external metacognitive bias. The same was found when raw metacognitive judgments rather than metacognitive bias scores were investigated. Moreover, group differences in metacognitive bias were observed not only immediately after the metacognitive training but also in a final judgement at the end of the experiment. Therefore, the effect of our metacognitive feedback training persisted beyond the initial manipulation phase. All group differences found were especially pronounced for participants' estimations of their internal memory abilities compared to their external memory abilities. This is consistent with the literature, as previous evidence suggests that individuals tend to be underconfident in their own, unaided memory abilities rather than underestimating the helpfulness of external tools [10, 18, 23, 24]. It is perhaps because metacognitive training appears to improve appraisals of memory, and underconfidence is especially pronounced for internal memory abilities, that the difference between groups is larger for the internal than the external bias. Altogether, the evidence of the present study shows that metacognitive training in the form of providing participants with feedback on their metacognitive judgments is effective in improving metacognitive judgment accuracy and in removing metacognitive bias. Similar effects of metacognitive training have been demonstrated in a study by Carpenter et al. [20], in which participants receiving feedback on their metacognitive judgements experienced increased metacognitive calibration relative to participants receiving feedback on task performance [see also: 19, 21, 25]. Further studies are needed to disentangle whether the benefit of metacognitive training was found due to the action of making predictions or whether participants must also receive feedback on these predictions, or even whether feedback alone can lead to such an effect.

The second line of investigation asked whether improved metacognitive accuracy leads to more optimal reminder setting, i.e. a reduction in reminder bias. Contrary to our predictions, both groups were significantly biased towards using more external reminders than was optimal. Despite improved metacognitive accuracy, the reminder bias was actually numerically larger for the feedback group than the control group. This suggests that the reminder bias cannot be fully explained by metacognitive error, seeing as it can be observed even when metacognitive bias is eliminated. An additional factor that may explain the bias towards reminders is a preference to avoid cognitive effort associated with use of internal memory [26–28]. According to the 'minimal memory' view, people have a general bias to use external information over internal memory representations [26]. This may be because cognitive effort is intrinsically costly [27, 28]. Consistent with this, recent evidence shows that the bias towards reminders is reduced (but not eliminated) when participants receive financial compensation based on the number of points they score, which is hypothesized to increase cognitive effort [29]. Seeing as the participants in the present study received a fixed payment, regardless of their performance, it remains to be seen whether metacognitive interventions might be effective under conditions of performance-based reward. We also note that participants underwent the forced trials before the choice trials in this experiment, unlike previous studies where the two types of trial have generally been intermixed [11, 29]. This could lead to inaccuracy in estimation of the reminder bias, if performance in the choice trials relative to forced trials was increased (e.g. due to practice) or reduced (e.g. due to fatigue). However, seeing as both feedback and control groups underwent the same procedure, this would affect both groups in the same manner. Therefore this issue does not confound the direct comparisons between groups, which were the main focus for the present study.

Offloading to reduce cognitive effort would be in line with recent evidence on pre-crastina-tion, in that individuals may want to complete a task sooner rather than later to reduce the effort of holding an intention in mind [30]. Another possibility is that participants may have preferred to use reminders in order to reduce variability in their performance, even if this resulted in a worse overall outcome when considering the mean level of performance [11]. Similarly, individuals may have chosen the external strategy to avoid "looking stupid" by mak-ing errors [31]. It is also possible that individuals chose a sub-optimal reminder strategy simply because they lack the arithmetic ability to weigh the two strategies properly, although it is not clear why this would cause systematic bias in one direction or the other. Despite the finding that reducing metacognitive bias did not reduce the reminder bias here, the present results do not rule out a metacognitive influence on cognitive offloading in other settings. Although the present experiment yielded a null effect, a previous study did demonstrate an effect of meta-cognitive interventions on the reminder bias [11; Experiment 3], showing that metacognitive interventions can influence reminder setting at least under certain circumstances. It is likely that any bias towards reminders is influenced by multiple factors, and the influence of factors such as cognitive effort does not rule out the influence of metacognitive factors as well. Indeed, a relationship between metacognitive bias and reminder bias was still observed in the control group of the present study, consistent with earlier findings [8–12].

Taking these results together, we suggest that metacognitive judgements play a role in the decision of whether to set external reminders, but other factors, such as avoidance of cognitive effort, may influence reminder setting too. Other cognitive offloading studies have also pro-posed that both metacognitive beliefs about expected performance as well as the effort required are critical in deciding whether to use an external resource [32].

To potentially reduce any preference for the avoidance of cognitive effort, future studies could provide a strong incentive, such as performance-based pay, for participants to behave optimally. Another method would be to amend the current study so that participants are merely asked which strategy option they would hypothetically choose in the choice trials, without having to use any cognitive effort in doing the actual task. This could help to establish the extent to which reminder-setting is influenced by avoidance of cognitive effort and meta-cognitive accuracy. Moreover, future work could explore item-by-item reminder selection, as this resembles how individuals offload memory in the real world: rather than offloading an entire block of information, they choose what information to offload on an item-to-item basis.

In conclusion, our feedback intervention was effective in improving participants metacog-nitive judgment accuracy. As this intervention improved judgment accuracy rather than increasing or decreasing confidence in memory abilities, it may be useful across multiple tasks or populations with different directions of bias. However, the extent to which metacognitive accuracy or other factors such as avoidance of cognitive effort influence the reminder bias remains uncertain. In order to promote the effective use of cognitive tools and to find inter-ventions that improve offloading behaviour, factors influencing the choice of reminder-setting must be further understood.

## Author Contributions

**Conceptualization:** Nicole C. Engeler, Sam J. Gilbert.

**Data curation:** Sam J. Gilbert.

**Formal analysis:** Nicole C. Engeler.

**Funding acquisition:** Sam J. Gilbert.

**Investigation:** Nicole C. Engeler, Sam J. Gilbert.

**Methodology:** Nicole C. Engeler, Sam J. Gilbert.

**Resources:** Sam J. Gilbert.

**Software:** Sam J. Gilbert.

**Supervision:** Sam J. Gilbert.

**Visualization:** Nicole C. Engeler.

**Writing – original draft:** Nicole C. Engeler.

**Writing – review & editing:** Nicole C. Engeler, Sam J. Gilbert.

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
