## [Decision Letter · Decision Letter 0]

9 Jul 2020

PONE-D-20-14992

The effect of metacognitive training on confidence and strategic reminder setting

PLOS ONE

Dear Dr. Engeler,

Thank you for submitting your manuscript to PLOS ONE. After careful consideration, we feel that it has merit but does not fully meet PLOS ONE’s publication criteria as it currently stands. Therefore, we invite you to submit a revised version of the manuscript that addresses the points raised during the review process.

I have now received three expert reviews and have also read the paper myself. As you can see, the comments are largely positive and I agree that there is much to like in this paper. To echo Reviewer 3, one laudable aspect is the preregistration of the analytic plan and the fact that the paradigm produces several metacognitive measures is also in itself a contribution to the literature.

Each reviewer has provided informative comments that I hope you will find useful. There are two types of key concerns that arose from the reviews. The first type of key concern is a relatively simple issue of clarifying the methods sections, to give provide more details about the online administration of the study and the task itself. I think part of the confusion might be the use of the word “trial” to refer to each block of 25 circle-drags (note that Reviewer 1’s comment 6 uses “trial” to refer to each circle drag; it was also a confusion that I had). Perhaps referring to each trial as a “block” might sidestep the issue.

The second type of key concern is about how exactly the data should be interpreted. For example, Reviewer 2 asked about the specific wording of the global metacognitive judgments questions. I suspect that one reason why they ask that question is to get at what participants specifically participants were being asked to make judgments about. For example, was the first global metacognitive judgment a prediction of how participants would perform on future trials or a postdiction of how well they did on the forced trials? Similarly, did participants interpret the second global judgment as a postdiction of how well they did in the choice trials? One reason this matter is that metacognitive bias (which is the DV on which effects are found) is based on the difference between the judgments and their actual performance on the forced trials. However, interpretation of this difference as over- or underconfidence is muddied if participants think they are making metacognitive judgment about anything other than performance on the forced trials. Another example is comment 5 by Reviewer 1 -- if participants are aware of flagging attention, this relatively  judgment could appear as a positive reminder bias. I do not expect that these are questions that can be definitively answered without a new study, but should at least be raised and addressed in the general discussion.

Finally, I had another more minor suggestion for the analysis of the relationship between reminder bias and metacognitive bias: currently, four correlations are reported and an interaction is implied. I would recommend considering using regression analyses to test the implied interaction (e.g., is the relationship between reminder bias and internal metacognitive bias moderated by group?).

I look forward to seeing the next iteration of this manuscript.

We look forward to receiving your revised manuscript.

Kind regards,

Veronica Yan, Ph.D.

Academic Editor

PLOS ONE

Journal Requirements:

Additional Editor Comments (if provided):

<accurate>

Reviewers' comments:

Reviewer's Responses to Questions</accurate>

**Comments to the Author**

1. Is the manuscript technically sound, and do the data support the conclusions?

Reviewer #1: Yes

Reviewer #2: Yes

Reviewer #3: Partly

2. Has the statistical analysis been performed appropriately and rigorously? 

Reviewer #1: Yes

Reviewer #2: Yes

Reviewer #3: Yes

3. Have the authors made all data underlying the findings in their manuscript fully available?

Reviewer #1: No

Reviewer #2: Yes

Reviewer #3: Yes

4. Is the manuscript presented in an intelligible fashion and written in standard English?

Reviewer #1: Yes

Reviewer #2: Yes

Reviewer #3: Yes

5. Review Comments to the Author

Reviewer #1: The authors examine whether a training period will shift metacognitive judgments about reminder-setting and adjust how many reminders learners choose to set. Learners move digital circles to assigned different sides of a box. Learners are randomly assigned to a metacognitive training condition in which they predict how many they will correctly move and receive feedback about their performance and predictions during forced trials OR to a baseline condition in which they do not practice metacognitive judgments. Practicing metacognitive judgments improves later metacognitive predictions but does not shift when and how they choose to use reminders.

I think cognitive offloading is an incredibly interesting and important area for investigation and understanding influences on it is worthwhile. This manuscript provides an incremental step towards greater understanding of cognitive offloading and reminder setting.

1. I wondering if practice making metacognitive judgments is truly a kind of “training.” The authors consider training to be making metacognitive predictions and getting feedback about their predictions. This does not fit my ideal of what training entails. Training seems like it would entail some kind of instruction. It may be more accurate to think of practice predictions as experience with the task, which then improve later predictions. Those that get experience making predictions become better at making predictions (and this improvement at the task continues throughout the experimental trials, even without feedback about estimates – i.e. the interaction of judgment time with other variables). Practice with this unusual or difficult task improves later performance at the task.

2. The primary conclusion (practice making metacognitive predictions does not change reminder setting) is evidenced by a null effect. The authors find no differences between conditions in reminder selections. This concern is somewhat allayed by differences in metacognitive monitoring accuracy, but null effects are not as powerful as significant differences. The authors suggest reasons why there may be null effects but cannot test their hypotheses. Understanding how to shift reminder setting and showing how to do so would be interesting future directions.

3. To calculate optimal selections, participants must compute a math equation – there is no other way to select optimally. Therefore, this “metacognitive” task may be more of an algebra problem than what is typically considered as metacognition. Succeeding at a math problem seems very different than on-going metacognitive monitoring. Do optimal selections in this task relate to math ability? How can learners choose optimally if they do not know how to solve the math problem?

4. I think that future directions could explore item-by-item reminder selections. Item-by-item selection processes could better mimic how learners choose to offload memory in the real world. Learners do not choose to offload an entire block of information; rather, they choose what items and information to offload on an item-by-item basis.

5. To calculate optimal allocation, the authors use performance on the forced trials at the beginning of the experiment. This could be inaccurate if learning occurs during training and performance improves. Alternatively, this could be inaccurate if fatigue and proactive interference happens and performance declines. In other words, the temporal separation of the forced trials from the choice trials could produce biases in calculating optimal selection. The procedure assumes that performance is static across the entire experiment, which seems unlikely. Further, we know attention waxes and wanes. If learners are sensitive to changes in their attention (as could be suggested by prior research, e.g., Markant, DuBrow, Davachi, and Gureckis, 2014), then they may be exercising better metacognitive control over reminder setting than can be picked up by the rough measurements in this procedure.

6. I think the authors need to be clearer about the 1-9 point assignment phases. It was unclear whether individual circles were assigned point values or whether the entire block of trials was assigned those point values (I figured out that it was the latter possibility, but it took a bit of time).

7. The authors need to be clearer about where they got their degrees of freedom around lines 279

8. The authors sometimes average across the two judgments and sometimes include the time point as a variable in the analysis. Can they be clearer about the differences in accounting for time point? And given the interactions with time point and other variables, does averaging across time point in some analyses obscure important differences?

9. I do not see a link to the data file used for this project.

Reviewer #2: I find this manuscript interesting, informative, and clearly written. I appreciate the authors’ work and can see why it would be of interest to those examining cognitive offloading. However, there are some potential areas for improvement, particularly in the motivation of the current work and the specification of the sample. I outline my suggestions below, in the order I noticed them in the paper.

Overall, I found that the authors could improve their situating of the current work, and describing why it is compelling – why would improving metacognitive judgment accuracy/reducing bias in offloading behavior be helpful? I think this motivation could be improved in both the Introduction and Discussion sections, perhaps using practical examples of why these factors matter. The larger issues at play here are interesting and could be enhanced in this manuscript.

---

The sample was collected via Amazon Mechanical Turk. Given the rather sizeable existing literature on the proper screening of MTurk participants, I think it's important to have additional detail about the sample.

How was age confirmed? Did you use Amazon MTurk's pre-screening age qualifications? To confirm that people were in the US as you stated, did you use some kind of IP address locator? Or were these items self-reported? If participants did self-report, how did you control for lying? (MTurk participants commonly respond to demographic questions based on their perceptions of demand characteristics, and are known to share those demand characteristics with their fellow MTurkers once they are discovered).

Some studies have reported that American MTurkers are more highly educated on average than other samples. Did you collect educational data, and if so, was that a factor (given the extant literature on the relationship between education and metacognition)? How about across age, as the sample ranges quite widely from 20-71 years old, and some literature suggests that younger and older adults differ in both their memory and metacognitive capabilities?

Did you use “catch questions” or “bot checks” to ensure participants were paying attention? If so, what were these items? When did they appear? If not, how did you verify that your sample did not contain bots?

Were participants screened to be fluent in English, or to have English as a first language? Among those whose first language was not English, how did you ensure fluency/understanding of the instructions, especially given that the task has multiple components?

Do the authors have a measure of how often Mturk participants left the page (e.g., to click to another window) when they were doing this task to assess potential distraction/lack of attention?

The authors report that “the experiment took approximately 45 minutes.” What was the standard deviation in time spent? Were data from any especially quick or especially slow outliers discarded? Could participants have clicked through very quickly (e.g., skimming through instructions), or were there minimum requirements for time spent on each page?

----

For conciseness, some repetitiveness between the Optimal Reminders Task section and the Procedure section could be decreased or consolidated.

Lines 236-237, please provide exact wording of the final metacognitive questions – was it some version of “how accurately do you think you performed on this task?”.

Finally, this may or may not be possible, but I think it would be interesting if readers could be directed to a quick video of the task in action in supplemental materials. Reading about it and seeing photos help, but a screen recording of the task could be even clearer and more compelling. If this exists elsewhere, directing readers toward it would be helpful.

Reviewer #3: The authors investigate behavior in a recently-developed metacognitive task in which participants drag a set of numbered circles to the bottom of the screen in order. Certain circles are initially marked in a different color and must be dragged to different locations, for which participants may optionally set an "external reminder." By manipulating the rewards & costs associated with setting a reminder, the paradigm allows the researchers how optimally people use external reminders. In the present study, the researchers additionally created an intervention in which experimental (but not control) participants made predictions and received feedback in the first block of trials. In the second block of trials, the experimental group showed more accurate performance predictions for both internal and external memory, but both groups still showed a behavioral bias towards setting more external reminders than optimal.

Overall, I found much to like about this manuscript.

One clear strength of the manuscript is the authors' commitment to open science. The authors have preregistered the participant exclusion criteria, experimental procedure and the analytic plan, and they have adhered to this closely. Among other laudable aspects of this plan, power analysis was used to set the target sample size a priori to avoid any risk of p-hacking.

Another strength is the data and analytic procedure. An exciting aspect of the paradigm is that it produces several measures, including actual performance, participants' bias in choosing external vs. internal memory, and metacognitive predictions, that each characterizes a separate aspect of behavior. The authors nicely outline each of these measures on p. 11 and clearly delineate the results and conclusions stemming from each one.

I did have some initial concerns about how strongly the results from this particular experimental paradigm support the claim of a general bias towards external reminders (with or without metacognitive training), but the authors generally acknowledge these concerns in their Discussion. First, participants might not be motivated to set an optimal indifference point simply because there is no real incentive to earn more points, but the authors acknowledge this in lines 389-394. An additional counter-explanation, which the authors may want to also discuss, is that participants use external reminders because they don't want to "look stupid" by making errors (Hawkins, Brown, Steyvers, & Wagenmakers, 2012), even if that results in a suboptimal payoff. Second, while it's also unclear whether the present intervention is effective because of the predictions, the feedback, or their combination, the authors similarly acknowledge this on lines 378-380 (although I think it would also be worthwhile to consider the possibility that feedback alone produces the effect, which the authors do not discuss). Thus, I find the authors' Discussion section generally effective.

But, there is the potential for one other major confound that was unclear to me when reading the manuscript. Lines 148-150 state that "when participants set an external reminder, target circles had to be dragged near the instructed alternative location." Am I to understand that offloading in this case involved moving the special-colored circles near or at their eventual target location? If so, I feel that this referring to this operation as just an a "external reminder" is a bit of a misnomer because putting the circle in the alternative location also helps to complete the eventual task. That could explain participants' bias to set reminders, since in doing so they are not only creating a reminder but also helping to complete the task ("pre-crastinating"; Rosenbaum et al., 2019). (Compare that to say, setting a reminder in one's phone to mow the lawn, in which case the phone alert serves as a reminder but does nothing to accomplish the lawn-mowing itself.) At the very least, the manuscript could make this issue more clear: Where, exactly, where the items moved when setting a reminder? And, if the items were indeed dragged to or near the target locations, I think the authors need to discuss how that may temper or alter their conclusions.

One other weakness of the current manuscript is that, while the empirical contributions are clear, the manuscript could do more to speak to the theoretical contributions. The theoretical motivation is a single paragraph in the introduction that briefly covers both the limits of working memory and the notion of offloading with technology. More contact with broader theory on external memory and metacognition could help contextualize the results. For example, can broader theoretical perspectives on metacognitive monitoring and control suggest anything about why people might have a reminder bias?

Nevertheless, I think this is a well-executed and well-analyzed study, and with some revision to the introduction and discussion, it will make a valuable contribution.

References

Hawkins, G. E., Brown, S. D., Steyvers, M., & Wagenmakers, E.-J. (2012). An optimal adjustment procedure to minimize experiment time in decisions with multiple alternatives. Psychonomic Bulletin & Review, 19, 339–348.

Rosenbaum, D, A., et al. (2019). Sooner rather than later: Precrastination rather than procrastination. Current Directions in Psychological Science, 28, 229-223.

6. PLOS authors have the option to publish the peer review history of their article (what does this mean?). If published, this will include your full peer review and any attached files.

Reviewer #1: No

Reviewer #2: No

Reviewer #3: No

---

## [Author Response · Author response to Decision Letter 0]

26 Aug 2020

We are very grateful to the editor and all three reviewers for their careful reading of our manuscript and their insightful comments, which we believe have greatly improved our manuscript. We have carefully revised the manuscript according to your comments, as summarised below. As well as uploading a clean version of the document as the main manuscript, we have also uploaded the version with track changes as a supplementary file so that all changes are clearly indicated.

Sincerely,

Sam Gilbert and Nicole Engeler 

Note: All line references refer to the “Manuscript with track changes” document. 

Editor comments

I have now received three expert reviews and have also read the paper myself. As you can see, the comments are largely positive and I agree that there is much to like in this paper. To echo Reviewer 3, one laudable aspect is the preregistration of the analytic plan and the fact that the paradigm produces several metacognitive measures is also in itself a contribution to the literature.

Thank you for this evaluation. 

Each reviewer has provided informative comments that I hope you will find useful. There are two types of key concerns that arose from the reviews. The first type of key concern is a relatively simple issue of clarifying the methods sections, to give provide more details about the online administration of the study and the task itself. I think part of the confusion might be the use of the word “trial” to refer to each block of 25 circle-drags (note that Reviewer 1’s comment 6 uses “trial” to refer to each circle drag; it was also a confusion that I had). Perhaps referring to each trial as a “block” might sidestep the issue.

Thank you for this summary. We have now clarified the methods, as explained in more detail below. We have also included a weblink (see lines 124-129) which allows anyone to try the experiment, exactly as it was presented to the actual participants. We hope this will further clarify any methodological points. We agree that our use of the word “trial” was potentially confusing, therefore we added a description of exactly what is meant by this on lines 62-63. This wording was chosen to be in line with previous studies (e.g. Gilbert et al., 2019). 

The second type of key concern is about how exactly the data should be interpreted. For example, Reviewer 2 asked about the specific wording of the global metacognitive judgments questions. I suspect that one reason why they ask that question is to get at what participants specifically participants were being asked to make judgments about. For example, was the first global metacognitive judgment a prediction of how participants would perform on future trials or a postdiction of how well they did on the forced trials? Similarly, did participants interpret the second global judgment as a postdiction of how well they did in the choice trials? One reason this matter is that metacognitive bias (which is the DV on which effects are found) is based on the difference between the judgments and their actual performance on the forced trials. However, interpretation of this difference as over- or underconfidence is muddied if participants think they are making metacognitive judgment about anything other than performance on the forced trials. Another example is comment 5 by Reviewer 1 -- if participants are aware of flagging attention, this relatively judgment could appear as a positive reminder bias. I do not expect that these are questions that can be definitively answered without a new study, but should at least be raised and addressed in the general discussion.

We agree that these are important issues and have clarified the manuscript accordingly. As well as providing a weblink to the full experimental task [lines 124-129] we have also added the exact wording for the 2nd global metacognitive judgments on lines 271-277. For all global metacognitive judgments, the wording made it clear that participants were judging their average ability for future trials rather than making postdictions. It was also made clear, using bold text, whether they were judging their ability for internal or external trials.

Regarding reviewer 1’s comment about the temporal separation between forced and choice trials, we agree that this is an important point. In most of our experiments using this paradigm we have intermixed force and choice trials in order to avoid this issue (e.g. Gilbert et al., 2020, Experiments 1 and 3). However, it was necessary to administer the forced trials first in this experiment seeing as this was where the metacognitive intervention took place. While we agree that this complicates interpretation of whether the reminder bias was positive or negative, we also note that this would apply equally to the feedback and no-feedback groups, therefore the direct comparison between the two groups – our main interest in this study - is not affected. We have added a discussion of this issue on lines 485-493. 

Finally, I had another more minor suggestion for the analysis of the relationship between reminder bias and metacognitive bias: currently, four correlations are reported and an interaction is implied. I would recommend considering using regression analyses to test the implied interaction (e.g., is the relationship between reminder bias and internal metacognitive bias moderated by group?).

We agree that in order to draw the conclusion that the relationship between internal metacognitive bias and reminder bias was specific to the no-feedback group, it would be necessary to demonstrate a significant difference between the correlations for the feedback and no-feedback groups. We now report this analysis on lines 405-409 and, seeing as it was not significant, we note that no strong conclusions may be drawn about specificity.

Reviewer 1

The authors examine whether a training period will shift metacognitive judgments about reminder-setting and adjust how many reminders learners choose to set. Learners move digital circles to assigned different sides of a box. Learners are randomly assigned to a metacognitive training condition in which they predict how many they will correctly move and receive feedback about their performance and predictions during forced trials OR to a baseline condition in which they do not practice metacognitive judgments. Practicing metacognitive judgments improves later metacognitive predictions but does not shift when and how they choose to use reminders.

I think cognitive offloading is an incredibly interesting and important area for investigation and understanding influences on it is worthwhile. This manuscript provides an incremental step towards greater understanding of cognitive offloading and reminder setting.

Thank you for this assessment.

1. I wondering if practice making metacognitive judgments is truly a kind of “training.” The authors consider training to be making metacognitive predictions and getting feedback about their predictions. This does not fit my ideal of what training entails. Training seems like it would entail some kind of instruction. It may be more accurate to think of practice predictions as experience with the task, which then improve later predictions. Those that get experience making predictions become better at making predictions (and this improvement at the task continues throughout the experimental trials, even without feedback about estimates – i.e. the interaction of judgment time with other variables). Practice with this unusual or difficult task improves later performance at the task.

Thank you, that is a good point. We agree that the wording choice of “training” can be confusing. However, we used this term in order to be consistent with the terminology previous studies have used (Carpenter et al., 2019). Therefore, we clarified what we mean by this term on lines 106-107.

2. The primary conclusion (practice making metacognitive predictions does not change reminder setting) is evidenced by a null effect. The authors find no differences between conditions in reminder selections. This concern is somewhat allayed by differences in metacognitive monitoring accuracy, but null effects are not as powerful as significant differences. The authors suggest reasons why there may be null effects but cannot test their hypotheses. Understanding how to shift reminder setting and showing how to do so would be interesting future directions.

We agree with these points, and have now included an additional section in the discussion (lines 505-507) where we acknowledge the null effect and discuss the relationship between metacognitive interventions and reminder-setting (as well as noting that such a relationship has previously been found in an earlier study).

3. To calculate optimal selections, participants must compute a math equation – there is no other way to select optimally. Therefore, this “metacognitive” task may be more of an algebra problem than what is typically considered as metacognition. Succeeding at a math problem seems very different than on-going metacognitive monitoring. Do optimal selections in this task relate to math ability? How can learners choose optimally if they do not know how to solve the math problem?

Thank you for this suggestion. We agree that participants may choose sub-optimally simply because they lack the arithmetic ability to weigh the two strategies properly, although it is not clear why this would cause systematic bias in one direction or the other. We now note this point in the discussion (lines 500-502).

4. I think that future directions could explore item-by-item reminder selections. Item-by-item selection processes could better mimic how learners choose to offload memory in the real world. Learners do not choose to offload an entire block of information; rather, they choose what items and information to offload on an item-by-item basis.

Thank you for another good point. We included this suggestion for future work in the discussion (line 535-536).

5. To calculate optimal allocation, the authors use performance on the forced trials at the beginning of the experiment. This could be inaccurate if learning occurs during training and performance improves. Alternatively, this could be inaccurate if fatigue and proactive interference happens and performance declines. In other words, the temporal separation of the forced trials from the choice trials could produce biases in calculating optimal selection. The procedure assumes that performance is static across the entire experiment, which seems unlikely. Further, we know attention waxes and wanes. If learners are sensitive to changes in their attention (as could be suggested by prior research, e.g., Markant, DuBrow, Davachi, and Gureckis, 2014), then they may be exercising better metacognitive control over reminder setting than can be picked up by the rough measurements in this procedure.

Thank you, this is a very valid concern. We now discuss this issue on lines 485-493. We note that previous studies have intermixed the forced and choice trials to avoid this problem. We also note that the issue would apply equally to both the feedback and the control groups, so should not confound any direct comparison between the groups. 

6. I think the authors need to be clearer about the 1-9 point assignment phases. It was unclear whether individual circles were assigned point values or whether the entire block of trials was assigned those point values (I figured out that it was the latter possibility, but it took a bit of time).

We have clarified the point assignment (line 185).

7. The authors need to be clearer about where they got their degrees of freedom around lines 279

Consistent with previous articles (e.g. Gilbert et al., 2020), degrees of freedom were based on t-tests conducted using R, without assuming equal variance. We now clarify this on lines (283-285). 

8. The authors sometimes average across the two judgments and sometimes include the time point as a variable in the analysis. Can they be clearer about the differences in accounting for time point? And given the interactions with time point and other variables, does averaging across time point in some analyses obscure important differences?

As determined in our pre-registration we first collapsed the two timepoints in order to conduct our main analyses of the metacognitive bias scores without inflating the type-1 error rate. This was followed by additional analyses separately investigating whether there was an effect of timepoint. We have now clarified this on lines 319-320. 

9. I do not see a link to the data file used for this project.

We have also uploaded data and code to https://osf.io/ebp4z/ so that all of the reported analyses can be reproduced. We now note this in lines 284-285. In reproducing all of our analyses using R, we noticed that there were some minor discrepancies and we also noticed an error in Table 1. All of these have been corrected. None of our conclusions were affected.

Reviewer 2

I find this manuscript interesting, informative, and clearly written. I appreciate the authors’ work and can see why it would be of interest to those examining cognitive offloading. However, there are some potential areas for improvement, particularly in the motivation of the current work and the specification of the sample. I outline my suggestions below, in the order I noticed them in the paper.

Overall, I found that the authors could improve their situating of the current work, and describing why it is compelling – why would improving metacognitive judgment accuracy/reducing bias in offloading behavior be helpful? I think this motivation could be improved in both the Introduction and Discussion sections, perhaps using practical examples of why these factors matter. The larger issues at play here are interesting and could be enhanced in this manuscript.

Thank you for this assessment. We expanded on the motivation of this manuscript on lines 117-120.

The sample was collected via Amazon Mechanical Turk. Given the rather sizeable existing literature on the proper screening of MTurk participants, I think it's important to have additional detail about the sample. How was age confirmed? Did you use Amazon MTurk's pre-screening age qualifications? To confirm that people were in the US as you stated, did you use some kind of IP address locator? Or were these items self-reported? If participants did self-report, how did you control for lying? (MTurk participants commonly respond to demographic questions based on their perceptions of demand characteristics, and are known to share those demand characteristics with their fellow MTurkers once they are discovered).

Age, gender and location were all self-reported, and we did not use MTurk’s pre-screening age qualifications, so we cannot exclude the possibility that some individuals gave false information. We now clarify this on lines 214 and 221.

Some studies have reported that American MTurkers are more highly educated on average than other samples. Did you collect educational data, and if so, was that a factor (given the extant literature on the relationship between education and metacognition)? How about across age, as the sample ranges quite widely from 20-71 years old, and some literature suggests that younger and older adults differ in both their memory and metacognitive capabilities?

We did not collect educational data, though we agree that it would be interesting to investigate the influence of this on offloading. We also agree that it is interesting to investigate potential age effects, but we did not do so here (although we have previously in Gilbert, 2015, Quarterly Journal of Experimental Psychology). This is because it is unclear how comparable younger and older mTurk participants are, and what the confounding variables might be. As an alternative, we have recently been investigating age effects in laboratory-based studies (Scarampi & Gilbert, under revision, Psychology and Aging: https://psyarxiv.com/vsa45/ ; Tsai, Kliegel, & Gilbert, in prep).

Did you use “catch questions” or “bot checks” to ensure participants were paying attention? If so, what were these items? When did they appear? If not, how did you verify that your sample did not contain bots?

We did not use catch questions or bot checks per se. However, seeing as participants needed to drag circles to specific parts of the screen in order to progress with the task, this would have excluded any attempts at automated responding. We also applied exclusion criteria to ensure that participants engaged with the task as intended, which we now note on lines 207-208. 

Were participants screened to be fluent in English, or to have English as a first language? Among those whose first language was not English, how did you ensure fluency/understanding of the instructions, especially given that the task has multiple components?

We included a practice session which required participants to respond accurately to a target circle in order to proceed, ensuring that participants understood the task instructions. This is now noted on lines 207-208. The full practice session can be sampled by visiting the demonstration weblink: http://samgilbert.net/demos/NE1/start.html

Do the authors have a measure of how often Mturk participants left the page (e.g., to click to another window) when they were doing this task to assess potential distraction/lack of attention?

Unfortunately we did not collect this measure, but we agree it would be useful information to collect in future work.

The authors report that “the experiment took approximately 45 minutes.” What was the standard deviation in time spent? Were data from any especially quick or especially slow outliers discarded? Could participants have clicked through very quickly (e.g., skimming through instructions), or were there minimum requirements for time spent on each page?

We have now included a more precise description of the experiment duration on lines 222-223: the median was 31 minutes. The standard deviation was large (30 minutes), however this is hard to interpret because it could be caused by participants loading up the first page of instructions in their web browser, then working on other tasks until returning to the browser tab later. It is common for mTurk workers to cue up several experiments in this manner before starting work. We did not impose any minimum requirements for time spent on each page, instead we applied exclusion criteria (line 210) to make sure that participants met a minimum level of accuracy when they performed the task. We also imposed a performance criterion on the practice session to ensure that participants understood the instructions (lines 224-226).

For conciseness, some repetitiveness between the Optimal Reminders Task section and the Procedure section could be decreased or consolidated.

Thank you for spotting the unnecessary repetition of information, we have consolidated this (lines 262-263).

Lines 236-237, please provide exact wording of the final metacognitive questions – was it some version of “how accurately do you think you performed on this task?”.

We have now added the exact wording of the final metacognitive judgments on lines 271-277.

Finally, this may or may not be possible, but I think it would be interesting if readers could be directed to a quick video of the task in action in supplemental materials. Reading about it and seeing photos help, but a screen recording of the task could be even clearer and more compelling. If this exists elsewhere, directing readers toward it would be helpful.

Thank you, this is a good idea. Rather than a video, we have provided a weblink (lines 124-129) so that anyone interested can complete the task themselves.

Reviewer 3

The authors investigate behavior in a recently-developed metacognitive task in which participants drag a set of numbered circles to the bottom of the screen in order. Certain circles are initially marked in a different color and must be dragged to different locations, for which participants may optionally set an "external reminder." By manipulating the rewards & costs associated with setting a reminder, the paradigm allows the researchers how optimally people use external reminders. In the present study, the researchers additionally created an intervention in which experimental (but not control) participants made predictions and received feedback in the first block of trials. In the second block of trials, the experimental group showed more accurate performance predictions for both internal and external memory, but both groups still showed a behavioral bias towards setting more external reminders than optimal.

Overall, I found much to like about this manuscript.

One clear strength of the manuscript is the authors' commitment to open science. The authors have preregistered the participant exclusion criteria, experimental procedure and the analytic plan, and they have adhered to this closely. Among other laudable aspects of this plan, power analysis was used to set the target sample size a priori to avoid any risk of p-hacking.

Another strength is the data and analytic procedure. An exciting aspect of the paradigm is that it produces several measures, including actual performance, participants' bias in choosing external vs. internal memory, and metacognitive predictions, that each characterizes a separate aspect of behavior. The authors nicely outline each of these measures on p. 11 and clearly delineate the results and conclusions stemming from each one.

Thank you for this evaluation.

I did have some initial concerns about how strongly the results from this particular experimental paradigm support the claim of a general bias towards external reminders (with or without metacognitive training), but the authors generally acknowledge these concerns in their Discussion. First, participants might not be motivated to set an optimal indifference point simply because there is no real incentive to earn more points, but the authors acknowledge this in lines 389-394. An additional counter-explanation, which the authors may want to also discuss, is that participants use external reminders because they don't want to "look stupid" by making errors (Hawkins, Brown, Steyvers, & Wagenmakers, 2012), even if that results in a suboptimal payoff. Second, while it's also unclear whether the present intervention is effective because of the predictions, the feedback, or their combination, the authors similarly acknowledge this on lines 378-380 (although I think it would also be worthwhile to consider the possibility that feedback alone produces the effect, which the authors do not discuss). Thus, I find the authors' Discussion section generally effective.

Thank you for the interesting additional points. We included the suggested possibility that participants use external reminders because they do not want to “look stupid” (lines 499-500), and that feedback alone could make the intervention effective (line 469).

But, there is the potential for one other major confound that was unclear to me when reading the manuscript. Lines 148-150 state that "when participants set an external reminder, target circles had to be dragged near the instructed alternative location." Am I to understand that offloading in this case involved moving the special-colored circles near or at their eventual target location? If so, I feel that this referring to this operation as just an a "external reminder" is a bit of a misnomer because putting the circle in the alternative location also helps to complete the eventual task. That could explain participants' bias to set reminders, since in doing so they are not only creating a reminder but also helping to complete the task ("pre-crastinating"; Rosenbaum et al., 2019). (Compare that to say, setting a reminder in one's phone to mow the lawn, in which case the phone alert serves as a reminder but does nothing to accomplish the lawn-mowing itself.) At the very least, the manuscript could make this issue more clear: Where, exactly, where the items moved when setting a reminder? And, if the items were indeed dragged to or near the target locations, I think the authors need to discuss how that may temper or alter their conclusions.

That is correct, offloading involved dragging target circles near their target locations in external trials. However, we do not believe that dragging target circles near their target location meaningfully reduces work, seeing as in either case the participant needs to make a small mouse movement to move a circle from one screen position to another. The physical demands of the two actions seem similar, i.e. a small movement of the wrist. Nevertheless, we agree that the literature on “pre-crastination” is highly relevant in this context and we now discuss it on lines 494-496.

One other weakness of the current manuscript is that, while the empirical contributions are clear, the manuscript could do more to speak to the theoretical contributions. The theoretical motivation is a single paragraph in the introduction that briefly covers both the limits of working memory and the notion of offloading with technology. More contact with broader theory on external memory and metacognition could help contextualize the results. For example, can broader theoretical perspectives on metacognitive monitoring and control suggest anything about why people might have a reminder bias?

We have now included some broader theoretical context on why people might exhibit a reminder bias in the discussion (lines 478-480).

Nevertheless, I think this is a well-executed and well-analyzed study, and with some revision to the introduction and discussion, it will make a valuable contribution.

Thank you for this assessment. 

References

Carpenter, J., Sherman, M. T., Kievit, R. A., Seth, A. K., Lau, H., & Fleming, S. M. (2019). Domain-general enhancements of metacognitive ability through adaptive training. Journal of Experimental Psychology: General, 148(1), 51.

Gilbert, S. J., Bird, A., Carpenter, J. M., Fleming, S. M., Sachdeva, C., & Tsai, P. C. (2019). Optimal use of reminders: Metacognition, effort, and cognitive offloading. Journal of Experimental Psychology: General.

Scarampi, C., & Gilbert, S. (2020, June 24). Age differences in strategic reminder setting and the compensatory role of metacognition. https://doi.org/10.31234/osf.io/vsa45

Scarampi, C., & Gilbert, S. J. (2020b). The effect of recent reminder setting on subsequent strategy and performance in a prospective memory task. Memory, 1-15.

---

## [Decision Letter · Decision Letter 1]

5 Oct 2020

The effect of metacognitive training on confidence and strategic reminder setting

PONE-D-20-14992R1

Dear Dr. Engeler,

We’re pleased to inform you that your manuscript has been judged scientifically suitable for publication and will be formally accepted for publication once it meets all outstanding technical requirements.

Kind regards,

Veronica Yan, Ph.D.

Academic Editor

PLOS ONE

Additional Editor Comments (optional):

Thank you for the submitted revision. As you can see, the reviewers felt that their comments were sufficiently addressed, and I agree. I do want to draw attention to one note that Reviewer 3 made about the consistency of the use of terms 'sequence' and 'trial' -- this is something that you may want to pay attention to for the final proofs.

Reviewers' comments:

Reviewer's Responses to Questions

**Comments to the Author**

1. If the authors have adequately addressed your comments raised in a previous round of review and you feel that this manuscript is now acceptable for publication, you may indicate that here to bypass the “Comments to the Author” section, enter your conflict of interest statement in the “Confidential to Editor” section, and submit your "Accept" recommendation.

Reviewer #1: All comments have been addressed

Reviewer #3: (No Response)

2. Is the manuscript technically sound, and do the data support the conclusions?

Reviewer #1: Yes

Reviewer #3: Yes

3. Has the statistical analysis been performed appropriately and rigorously? 

Reviewer #1: Yes

Reviewer #3: Yes

4. Have the authors made all data underlying the findings in their manuscript fully available?

Reviewer #1: Yes

Reviewer #3: Yes

5. Is the manuscript presented in an intelligible fashion and written in standard English?

Reviewer #1: Yes

Reviewer #3: Yes

6. Review Comments to the Author

Reviewer #1: My comments were adequately addressed.

I think this manuscript will make an interesting, incremental addition to the literature on metacognition.

Reviewer #3: I was Reviewer #3 on the previous version of the manuscript. In my initial review, I expressed enthusiasm about the authors' experimental paradigm and its ability to yield multiple informative measures relevant to metacognition. I also appreciated the authors' commitment to open-science practices.

My concerns had been relatively small. I had expressed some concern that, if setting reminders involved moving the special-colored circles nearer to their eventual target location, participants might doing so in order to partially solve the eventual task ("pre-crastination"), in addition to acting as a reminder per se. I also noted another explanation of the reminder bias is that participants might want to avoid "looking stupid" by making errors, even if that results in a suboptimal payout. Lastly, I felt that the authors could do more to situate the work within broader theory.

I am pleased to report that the revision has addressed all of my concerns.

The authors have clarified the methodological details that I and the other reviewers have asked about; in particular, the addition of a link to a demo version of the experiment is a fantastic decision that I hope to see implemented in more papers in the future.

The authors also now use their Discussion section to acknowledge several potential limitations and counter-explanations that I and the other review mentioned. What I think is particularly encouraging about these revisions is that the authors have tied these other factors back into their "minimal-memory" theoretical account; for instance, pre-crastination may actually be part of the explanation as to *why* there is a reminder bias insofar as removing an intention from mind saves mental effort. That is, not only have the authors acknowledged this possibility, but have made a persuasive argument that it in fact advances their theoretical view. Fantastic.

One small comment (which should not preclude acceptance of the manuscript): Based on comments from the editor, the authors have clarified that one "trial" refers to the entire sequence of 25 circles. That is fine with me; however, I notice that the alternate term "sequence" is used in several places to refer to an entire trial (e.g., lines 146, 151, 157). In their final submission, the authors may want to change these usages to "trial" as well so that a reader is not misled into thinking that there is a difference between a "trial" and a "sequence".

7. PLOS authors have the option to publish the peer review history of their article (what does this mean?). If published, this will include your full peer review and any attached files.

Reviewer #1: No

Reviewer #3: No